# Unsupervised learning of features and object boundaries from local prediction

## Abstract

A visual system has to learn both which features to extract from images and how to group locations into (proto-)objects. Those two aspects are usually dealt with separately, although predictability is discussed as a cue for both. To incorporate features and boundaries into the same model, we model a layer of feature maps with a pairwise Markov random field model in which each factor is paired with an additional binary variable, which switches the factor on or off. Using one of two contrastive learning objectives, we can learn both the features and the parameters of the Markov random field factors from images without further supervision signals. The features learned by shallow neural networks based on this loss are local averages, opponent colors, and Gabor-like stripe patterns. Furthermore, we can infer connectivity between locations by inferring the switch variables. Contours inferred from this connectivity perform quite well on the Berkeley segmentation database (BSDS500) without any training on contours. Thus, computing predictions across space aids both segmentation and feature learning, and models trained to optimize these predictions show similarities to the human visual system. We speculate that retinotopic visual cortex might implement such predictions over space through lateral connections.

## 1 Introduction

A long-standing question about human vision is how representations initially be based on parallel processing of retinotopic feature maps can represent *objects* in a useful way. Most research on this topic has focused on computing later object-centered representations from the feature map representations. Psychology and neuroscience identified features that lead to objects being grouped together [37, 38], established feature integration into coherent objects as a sequential process [73], and developed solutions to the binding problem, i.e. ways how neurons could signal whether they represent parts of the same object [17, 57, 67, 72]. In computer vision, researchers also focused on how feature map representations could be turned into segmentations and object masks. Classically, segmentation algorithm were clustering algorithms operating on extracted feature spaces [2, 12, 13, 16, 66], and this approach is still explored with more complex mixture models today [74]. Since the advent of deep neural network models, the focus has shifted towards models that directly map to contour maps or semantic segmentation maps [21, 27, 39, 50, 65, 83], as reviewed in [54].

Diverse findings suggest that processing within the feature maps take object boundaries into account. For example, neurons appear to encode border ownership [34, 57, 63] and to fill in information across surfaces [40] and along illusory contours [23, 76]. Also, attention spreading through the feature maps seems to respect object boundaries [4, 59]. And selecting neurons that correspond to an object takes time, which scales with the distance between the points to be compared [35, 41]. Finally, a long history of psychophysical studies showed that changes in spatial frequency and orientation

content can define (texture) boundaries [e.g. 5, 45, 81]. In both human vision and computer vision, relatively little attention has been given to these effects of grouping or segmentation on the feature maps themselves.

Additionally, most theories for grouping and segmentation take the features in the original feature maps as given. In human vision, these features are traditionally chosen by the experimenter [37, 73, 72] or are inferred based on other research [57, 63]. Similarly, computer vision algorithms used off-the-shelf feature banks originally [2, 12, 13, 16, 66], and have recently moved towards deep neural network representations trained for other tasks as a source for feature maps [21, 27, 39, 50, 65, 83].

Interestingly, predictability of visual inputs over space and time has been discussed as a solution for both these limitations of earlier theories. Predictability has been used as a cue for segmentation since the law of common fate of Gestalt psychology [37], and both lateral interactions in visual cortices and contour integration respect the statistics of natural scenes [19, 20]. Among other signals like sparsity [55] or reconstruction [36], predictability is also a well known signal for self-supervised learning of features [80], which has been exploited by many recent contrastive learning [e.g. 15, 24, 29, 75] and predictive coding schemes [e.g. 51, 52, 75] for self-supervised learning. However, these uses of predictability for feature learning and for segmentation are usually studied separately.

Here, we propose a model that learns both features and segmentation without supervision. Predictions between locations provide a self-supervised loss to learn the features, how to perform the prediction and how to infer which locations should be grouped. Also, this view combines contrastive learning [24, 75], a Markov random field model for the feature maps [46] and segmentation into a coherent framework. We implement our model using some shallow architectures. The learned features resemble early cortical responses and the object boundaries we infer from predictability align well with human object contour reports from the Berkeley segmentation database (BSDS500 [2]). Thus, retinotopic visual cortex might implement similar computational principles as we propose here.

## 2 Model

To explain our combined model of feature maps and their local segmentation information, we start with a Gaussian Markov random field model [46] with pairwise factors. We then add a variable $w \in \{0, 1\}$ to each factor that governs whether the factor enters the product or not. This yields a joint distribution for the whole feature map and all $w$'s. Marginalizing out the $w$'s yields a Markov random field with "robust" factors for the feature map, which we can use to predict feature vectors from the vectors at neighboring positions. We find two contrastive losses based on these predictions that can be used to optimize the feature extraction and the factors in the Markov random field model.

We model the distribution of $k$-dimensional feature maps $\mathbf{f} \in \mathbb{R}^{k,m',n'}$ that are computed from input images $I \in \mathbb{R}^{c,m,n}$ with $c = 3$ color channels (see Fig. 1 A & B). We use a Markov random field model with pairwise factors, i.e. we define the probability of encountering a feature map $\mathbf{f}$ with entries $f_i$ at locations $i \in [1 \dots m'] \times [1 \dots n']$ as follows:

$$p(\mathbf{f}) \propto \prod_i \psi_i(f_i) \prod_{(i,j) \in N} \psi_{ij}(f_i, f_j), \tag{1}$$

where $\psi_i$ is the local factor, $N$ is the set of all neighboring pairs, and $\psi_{ij}$ is the pairwise factor between positions $i$ and $j$[1]. We will additionally assume shift invariance, i.e. each point has the same set of nearby relative positions in the map as neighbors, $\psi_i$ is the same factor for each position, and each factor $\psi_{ij}$ depends only on the relative position of $i$ and $j$.

We now add a binary variable $w \in \{0, 1\}$ to each pairwise factor that encodes whether the factor is 'active' ($w = 1$) for that particular image (Fig. 1 C). To scale the probability of $w = 1$ and $w = 0$ relative to each other, we add a factor that scales them with constants $p_{ij} \in [0, 1]$ and $1 - p_{ij}$ respectively:

$$p(\mathbf{f}, \mathbf{w}) \propto \prod_i \psi_i(f_i) \prod_{(i,j) \in N} p_{ij}^{w_{ij}} (1 - p_{ij})^{1-w_{ij}} \psi_{ij}(f_i, f_j)^{w_{ij}} \tag{2}$$

---

[1]$i$ and $j$ thus have two entries each

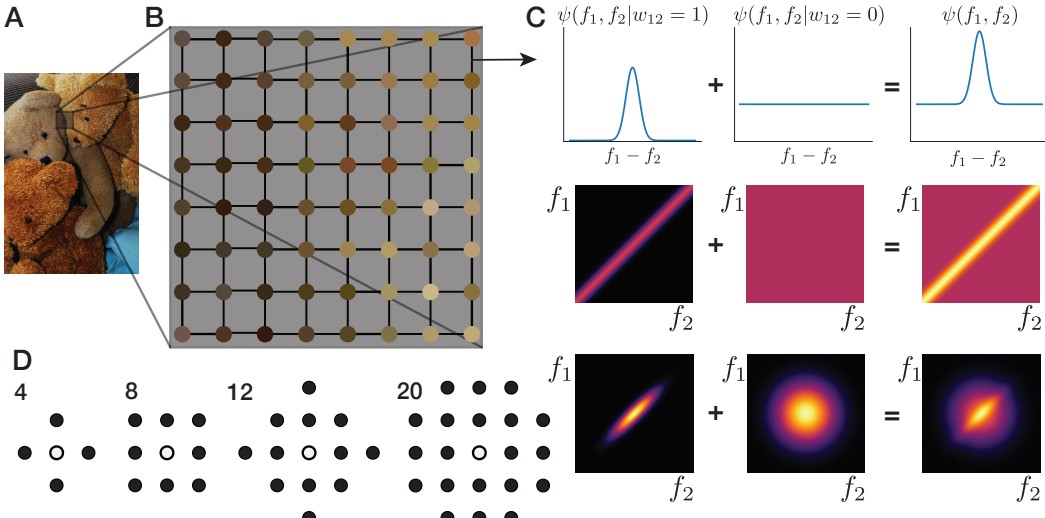

Figure 1: Illustration of our Markov random field model for the feature maps. **A**: An example input image. **B**: Feature map with 4 neighborhood connectivity and pixel color as the extracted feature. In the actual models, these feature maps are higher dimensional maps extracted by a convolutional neural network model. **C**: Illustration of the factor that links the feature vectors at two neighboring locations for a 1D feature. Top row: projection of the factor $\psi_{ij}$ onto the difference between the features value $f_i - f_j$, showing the combination of a Gaussian around 0 and a constant function for the connection variable $w_{ij}$ being 1 or 0 respectively. Middle row: 2D representation of the factor and its parts plotted against both feature values. Bottom row: Multiplication of the middle row with the standard normal factor for each position yielding the joint distribution of two isolated positions. **D**: Neighborhoods of different sizes used in the models, scaling from 4 to 20 neighbors for each location.

Finally, we assume that the factors are Gaussian and the feature vectors are originally normalized to have mean 0 and variance 1:

$$p(\mathbf{f}, \mathbf{w}) = \frac{1}{Z_0} \mathcal{N}(\mathbf{f}, 0, \mathbf{I}) \prod_{(i,j) \in N} \frac{p_{ij}^{w_{ij}} (1 - p_{ij})^{1 - w_{ij}}}{Z(w_{ij}, C_{ij})} \exp\left(-\frac{w_{ij}}{2}(f_i - f_j)^T C_{ij}(f_i - f_j)\right), \quad (3)$$

where $Z_0$ is the overall normalization constant, $N(\mathbf{f}, 0, \mathbf{I})$ is the density of a standard normal distribution with $k \times m' \times n'$ dimensions, $C_{ij}$ governs the strength of the coupling in the form of a precision matrix, which we will assume to be diagonal, and $Z(w_{ij}, C_{ij})$ scales the distributions with $w_{ij} = 0$ and $w_{ij} = 1$ relative to each other.

We set $Z(w_{ij}, C_{ij})$ to the normalization constant of the Gaussian with standard Gaussian factors for $f_i$ and $f_j$ respectively. For $w = 0$ this is just $(2\pi)^{-k}$, the normalization constant of a standard Gaussian in $2k$ dimensions. For $w = 1$ we get:

$$Z(w_{ij} = 1, C_{ij}) = \int \int \exp\left(-\frac{1}{2} f_i^T f_i - \frac{1}{2} f_j^T f_j - \frac{1}{2}(f_i - f_j)^T C_{ij}(f_i - f_j)\right) df_i df_j \quad (4)$$

$$= (2\pi)^{-k} \det \begin{vmatrix} I + C_{ij} & C_{ij} \\ C_{ij} & I + C_{ij} \end{vmatrix}^{\frac{1}{2}} \quad (5)$$

$$= (2\pi)^{-k} \prod_l \sqrt{1 + 2c_{ll}} \quad (6)$$

which we get by computing the normalization constant of a Gaussian with the given precision and then using the assumption that $C_{ij}$ is a diagonal matrix with diagonal entries $c_{ll}$.

This normalization depends only on $w$ and the coupling matrix $C$ of the factor $\psi_{ij}$ and thus induces a valid probability distribution on the feature maps. Two points are notable about this normalization though: First, once other factors also constrain $f_i$ and/or $f_j$, this normalization will not guarantee $p(w_{ij} = 1) = p_{ij}$.[2] Second, the $w_{ij}$ are not independent in the resulting distribution. For example, if pairwise factors connect $a$ to $b$, $b$ to $c$ and $a$ to $c$ the corresponding $w$ are dependent, because $w_{ab} = 1$ and $w_{bc} = 1$ already imply a smaller difference between $f_a$ and $f_c$ than if these factor were inactive, which increases the probability for $w_{ac} = 1$.

## 2.1 Learning

To learn our model from data, we use a contrastive learning objective on the marginal likelihood $p(\mathbf{f})$. To do so, we first need to marginalize out the $w$'s, which is fortunately simple, because each $w$ affects only a single factor:

$$p(\mathbf{f}) = \sum_{\mathbf{w}} p(\mathbf{f}, \mathbf{w}) = \frac{1}{Z_0} \mathcal{N}(\mathbf{f}, 0, \mathbf{I}) \prod_{(i,j) \in N} [p_{ij} \psi_{ij}(f_i, f_j) + (1 - p_{ij})] \tag{7}$$

Using this marginal likelihood directly for fitting is infeasible though, because computing $Z_0$, i.e. normalizing this distribution is not computationally tractable.

We resort to contrastive learning to fit the unnormalized probability distribution [24], i.e. we optimize discrimination from a noise distribution with the same support as the target distribution. Following [75] we do not optimize the Markov random field directly, but optimize predictions based on the model using features from other locations as the noise distribution. For this noise distribution, the factors that depend only on a single location (the first product in (1)) will cancel. We thus ignore the $N(\mathbf{f}, 0, \mathbf{I})$ in our optimization and instead normalize the feature maps to mean 0 and unit variance across each image. We define two alternative losses that make predictions for positions based on all their neighbors or for a single factor respectively.

### 2.1.1 Position loss

The *position loss* optimizes the probability of the feature vector at each location relative to the probability of randomly chosen other feature vectors from different locations and images:

$$l_{\text{pos}}(\mathbf{f}) = \sum_i \log \frac{p(f_i | f_j \forall j \in N(i))}{\sum_{i'} p(f_{i'} | f_j \forall j \in N(i))} \tag{8}$$

$$= \sum_i \sum_{j \in N(i)} \log \psi_{ij}(f_i, f_j) - \sum_i \log \left( \sum_{i'} \exp \left[ \sum_{j \in N(i)} \log \psi_{ij}(f_{i'}, f_j) \right] \right), \tag{9}$$

where $N(i)$ is the set of neighbors of $i$.

This loss is consistent with the prediction made by the whole Markov random field, but is relatively inefficient, because the predicted distribution $p(f_i | f_j \forall j \in N(i))$ and the normalization constants for these conditional distributions are different for every location $i$. Thus, the second term in equation (9) cannot be reused across the locations $i$. Instead, we need to compute the second term for each location separately, which requires a similar amount of memory as the whole feature representation for each negative sample $i'$ and each neighbor.

To enable a sufficiently large set of negative points $i'$ with the available memory, we compute this loss multiple times with few negative samples and sum the gradients. This trick saves memory, because we can free the memory for the loss computation after each repetition. As the initial computation of the feature maps is the same for all negative samples, we can save some computation for this procedure by computing the feature maps only once. To propagate the gradients through this single computation, we add up the gradients of the loss repetitions with regard to the feature maps and then propagate this summed gradient through the feature map computation. This procedure does not save computation time compared to the loss with many negative samples, as we still need to calculate the evaluation for each position and each sample in the normalization set.

---

[2]Instead, $p(w_{ij} = 1)$ will be higher, because other factors increase the precision for the feature vectors, which makes the normalization constants more similar.

 **2.1.2 Factor loss**

 The *factor loss* instead maximizes each individual factor for the correct feature vectors relative to
 random pairs of feature vectors sampled from different locations and images:

$$l_{\text{fact}} = \sum_{i,j} \log \frac{\psi_{ij}(f_i, f_j)}{\sum_{i',j'} \psi_{ij}(f_{i'}, f_{j'})} \tag{10}$$

$$= \sum_{i,j} \log \psi_{ij}(f_i, f_j) - \sum_{i,j} \log \sum_{i',j'} \psi_{ij}(f_{i'}, f_{j'}), \tag{11}$$

 where $i, j$ index the correct locations and $i', j'$ index randomly drawn locations, in our implementation
 generated by shuffling the feature maps and taking all pairs that occur in these shuffled maps.

 This loss does not lead to a consistent estimation of the MRF model, because the prediction $p(f_i|f_j)$
 should not be based only on the factor $\psi_{ij}$, but should include indirect effects as $f_j$ also constrains
 the other neighbors of $i$. Optimizing each factor separately will thus overaccount for information
 that could be implemented in two factors. However, this loss has the distinct advantage that the same
 noise evaluations can be used for all positions and images in a minibatch, which enables a much
 larger number of noise samples and thus much faster convergence.

 **2.1.3 Optimization**

 We optimize all weights of the neural network used for feature extraction and the parameters of the
 random field, i.e. the connectivity matrices $C$ and the $p_{ij}$ for the different relative spatial locations
 simultaneously. As an optimization algorithm we use stochastic gradient descent with momentum.
 Further details of the optimization can be found in the supplementary materials.

 **2.2 Segmentation inference**

 Computing the probability for any individual pair of locations $(i, j)$ to be connected, i.e. computing
 $p(w_{ij} = 1|\mathbf{f})$, depends only on the two connected feature vectors $f_i$ and $f_j$:

$$\frac{p(w_{ij} = 1|\mathbf{f})}{p(w_{ij} = 0|\mathbf{f})} = \frac{p_{ij}}{(1 - p_{ij})} \frac{Z(w_{ij} = 0, C_{ij})}{Z(w_{ij} = 1, C_{ij})} \exp\left(-(f_i - f_j)^T C_{ij}(f_i - f_j)\right) \tag{12}$$

 This inference effectively yields a connectivity measure for each pair of neighboring locations, i.e. a
 sparse connectivity matrix. Given that we did not apply any prior information enforcing continuous
 objects or contours, the inferred $w_{ij}$ do not necessarily correspond to a valid segmentation or set of
 contours. Finding the best fitting contours or segmentation for given probabilities for the $w$s is an
 additional process, which in humans appears to be an attention-dependent serial process [35, 63].

 To evaluate the detected boundaries in computer vision benchmarks, we nonetheless need to convert
 the connectivity matrix we extracted into a contour image. To do so, we use the spectral-clustering-
 based globalization method developed by [2]. This method requires that all connection weights
 between nodes are positive. To achieve this, we transform the log-probability ratios for the $w_{ij}$ as
 follows: For each image, we find the $30\%$ quantile of the values, subtract it from all log-probability
 ratios, and set all values below $0.01$ to $0.01$. We then compute the smallest eigenvectors of the graph
 Laplacian as in graph spectral clustering. These eigenvectors are then transformed back into image
 space and are filtered with simple edge detectors to find the final contours.

 # 3 Evaluation

 We implement 3 model types implementing feature extractions of increasing complexity in PyTorch
 [56]:

 **Pixel value model.** For illustrative purposes, we first apply our ideas to the rgb pixel values of an
 image as features. This provides us with an example, where we can easily show the feature values
 and connections. Additionally, this model provides an easy benchmark for all evaluations.

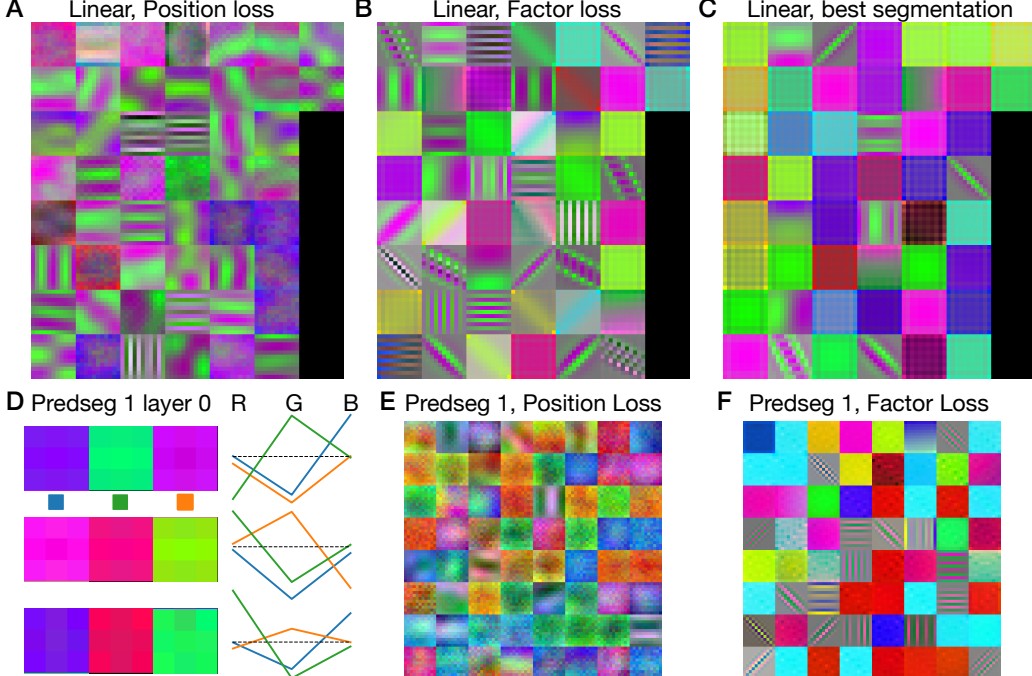

Figure 2: Example linear filter weights learned by our models. Each individual filter is normalized to minimum 0 and maximum 1. As weights can be negative even a zero weight can lead to a pixel having some brightness. For example, a number of channels load similarly on red and green across positions. Where these weights are positive the filter appears yellow and where the weights are negative filter appears blue, even if the blue channel has a zero weight. **A-C**: Feature weights learned by the linear model. **A**: Using the position loss. **B**: Using the factor loss. **C**: The weights of the model that leads to the best segmentation performance, i.e. the one shown in Figure 3. **D**: Weights of the first convolution in predseg1. Next to the filter shapes, which are nearly constant, we plot the average weight of each channel onto the three color channels of the image. **E** Predseg1 filters in the second convolution for a network trained with the position based loss. **F**: Predseg1 filters in the second convolution for a network trained with the factor based loss.

**Linear model.** As the simplest kind of model that allows learning features, we use a single convolutional deep neural network layer as our feature model. Here, we use 50 $11 \times 11$ linear features.

**Predseg1**: To show that our methods work for more complex architecture with non-linearities, we use a relatively small deep neural network with 4 layers (2 convolutional layers and 2 residual blocks with subsampling layers between them, see supplement for details).

For each of these architectures, we train 24 different networks with all combinations of the following settings: 4 different sizes of neighborhoods (4, 8, 12, or 20 neighbors, see Fig. 1D); 3 different noise levels (0, 0.1, 0.2) and the two learning objectives. As a training set, we used the unlabeled image set from MS COCO [48], which contains 123,404 color images with varying resolution. To enable batch processing, we randomly crop these images to $256 \times 256$ pixel resolution, but use no other data augmentation (See supplementary information for further training details).

We want to evaluate whether our models learn meaningful features and segmentations. To do so, we first analyze the features in the first layers of our networks where we can judge whether features are representative of biological visual systems. In particular, we extract segmentations from our activations and evaluate those on the Berkeley Segmentation Dataset [2, BSDS500]

## 3.1 Learned features

**Linear Model** We first analyze the weights in our linear models (Fig 2 A-C). All instances learn local averages and Gabor-like striped features, i.e. spatial frequency and orientation tuned features

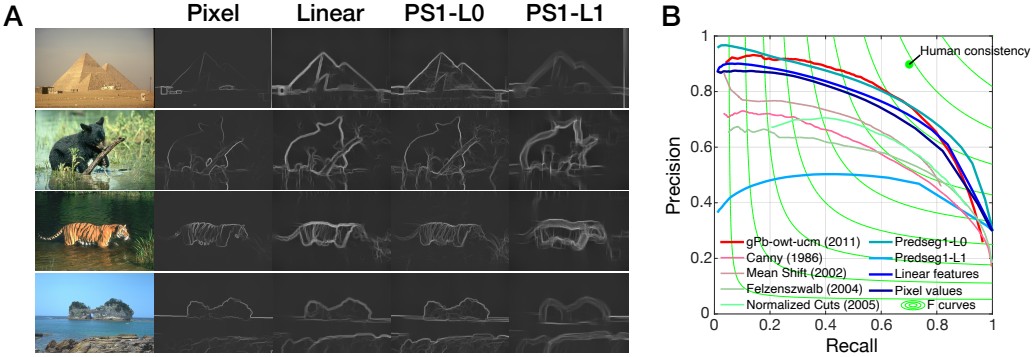

Figure 3: Contour detection results. **A**: Example segmentations from our models. **B**: Precision-recall curves for our models on the Berkeley segmentation dataset, with some other models for comparison as evaluated by [2]: gPb-uwt-ucm, the final algorithm combining all improvements [2], Canny's classical edge detector [7], the mean shift algorithm [12], Felzenschwalbs algorithm [16] and segmentation based on normalized cuts [13]. For all comparison algorithms evaluations on BSDS were extracted from the figure by [2]

188 with limited spatial extend. These features clearly resemble receptive fields of neurons in primary
189 visual cortex. Additionally, there appears to be some preference for features that weight the red and
190 green color channels much stronger than the blue channel, similar to the human luminance channel,
191 which leads to the yellow-blue contrasts in the plots. There is some difference between the two
192 learning objectives though. The position based loss generally leads to lower frequency and somewhat
193 noisier features. This could either be due to the higher learning efficiency of the factor based loss, i.e.
194 the factor based loss is closer to convergence, or due to a genuinely different optimization goal.

195 **Predseg1** In Predseg1, we first analyze the layer 0 convolution (Fig. 2D), which has only 3 channels
196 with $3 \times 3$ receptive fields, which we originally introduced as a learnable downsampling. This layer
197 consistently converges to applying near constant weights over space. Additionally, exactly one of
198 the channels has a non-zero mean (the 3rd, 1st and 3rd in Fig. 2D) and the other two take balanced
199 differences between two of the channels (red vs green and green vs. blue in the examples). This
200 parallels the luminance and opponent color channels of human visual perception.

201 In the second convolution, we observe a similar pattern of oriented filters and local averages as in
202 the linear model albeit in false color as the input channels are rotated by the weighting of the layer 0
203 convolution (Fig. 2 E & F).

## 3.2 Contour detection

205 To evaluate whether the connectivity information extracted by our model corresponds to human
206 perceived segmentation, we extract contours from our models and compare them to contours reported
207 by humans for the Berkeley Segmentation database [2, 53]. This database contains human drawn
208 object boundaries for 500 natural images and is accompanied by methods for evaluating segmentation
209 models. Using the methods provided with the database, we compute precision-recall curves for each
210 model and use the best F-value (geometric mean of precision and recall) as the final evaluation metric.

211 As we had multiple models to choose from, we choose the models from each class that perform
212 best on the *training data* for our reports. For all models this was one of the models with the largest
213 neighborhood, i.e. using 20 neighbors, and the factor loss. It seems the factor loss performed
214 better simply due to its technical efficiency advantage as discussed above. Performance increases
215 monotonically with neighborhood size and Markov random field based approaches to semantic
216 segmentation also increased their performance with larger neighborhoods up to fully connected
217 Markov random fields [43, 8, 9]. We thus expect that larger neighborhoods could work even better.

218 Qualitatively, we observe that all our models yield sensible contour maps (see Fig. 3 A). Even the
219 contours extracted from the pixel model yield sensible contours. Additionally, we note that the linear
220 model and Layer 1 of the predseg model tend to produce double contours, i.e. they tend to produce

Table 1: Numerical evaluation for various algorithms on the BSDS500 dataset. Precision and recall are only given for ODS, i.e. with a the threshold fixed across the whole dataset.

| model | Recall | Precision | F(ODS) | F(OIS) | Area_PR |
|---|---|---|---|---|---|
| Deep Contour** [65] | – | – | 0.76 | 0.78 | 0.80 |
| HED** [83] | – | – | 0.79 | 0.81 | 0.84 |
| RCF** [50] | – | – | 0.81 | 0.83 | – |
| Deep Boundary** [39] | – | – | 0.813 | 0.831 | 0.866 |
| BDCN** [27] | – | – | 0.83 | 0.84 | 0.89 |
| Canny* [7] | – | – | 0.60 | 0.63 | 0.58 |
| Mean Shift* [12] | – | – | 0.64 | 0.68 | 0.56 |
| Felzenszwalb* [16] | – | – | 0.61 | 0.64 | 0.56 |
| Normalized Cuts* [13] | – | – | 0.64 | 0.68 | 0.45 |
| gPb-owt-ucm [2] | 0.73 | 0.73 | 0.73 | 0.76 | 0.73 |
| Pixel | 0.73 | 0.66 | 0.69 | 0.69 | 0.73 |
| linear | 0.78 | 0.66 | 0.72 | 0.73 | 0.75 |
| Predseg1-Layer 0 | 0.79 | 0.69 | 0.74 | 0.73 | 0.80 |
| Predseg1-Layer 1 | 0.74 | 0.47 | 0.57 | 0.59 | 0.45 |

*: Evaluation of these algorithms taken from [2]. **: Supervised DNNs, evaluation taken from [27].

two contours on either side of the contour reported by human subjects with some area between them connected to neither side of the contour.

Quantitatively, our models also perform well except for the deeper layers of Predseg 1 (Fig. 3B and Table 1). The other models beat most hand-crafted contour detection algorithms that were tested on this benchmark [7, 12, 13, 16] and perform close to the gPb-owt-ucm contour detection and segmentation algorithm [2] that was the state of the art at the time. Layer-0 of Predseg 1 performs best followed by the linear feature model and finally the pixel value model. Interestingly, the best performing models seem to be mostly the local averaging models (cf. Fig. 2 C). In particular, the high performance of the first layer of Predseg 1 is surprising, because it uses only $3 \times 3$ pixel local color averages as features.

Since the advent of deep neural network models, networks trained to optimize performance on image segmentation have reached much higher performance on the BSDS500 benchmark, essentially reaching perfect performance up to human inconsistency [e.g. 27, 39, 49, 50, 65, 71, 83, see Table 1]. However, these models all require direct training on human reported contours and often use features learned for other tasks. There are also a few deep neural network models that attempt unsupervised segmentation [e.g. 10, 47, 82], but we were unable to find any that were evaluated on the contour task of BSD500. The closest is perhaps the W-net [82], which used an autoencoder structure with additional constraints and was evaluated on the segmentation task on BSDS500 performing slightly better than gPb-owt-ucm.

## 4 Discussion

We present a model that can learn features and local segmentation information from images without further supervision signals. This model integrates the prediction task used for feature learning and the segmentation task into the same coherent probabilistic framework. This framework and the dual use for the connectivity information make it seem sensible to represent this information. Furthermore, the features learned by our models resemble receptive fields in the retina and primary visual cortex and the contours we extract from connectivity information match contours drawn by human subject fairly well, both without any training towards making them more human-like.

To improve biological plausibility, all computations in our model are local and all units are connected to the same small, local set of other units throughout learning and inference, which matches early visual cortex, in which the lateral connections that follow natural image statistics are implemented anatomically [6, 31, 59, 70]. This in contrast to other ideas that require flexible pointers to arbitrary locations and features [as discussed by 64] or capsules that flexibly encode different parts of the input [14, 42, 61, 62]. Nonetheless, we employ contrastive learning objectives and backpropagation here,

for which we do not provide a biologically plausible implementations. However, there is currently active research towards biologically plausible alternatives to these algorithms [e.g. 32, 84].

Selecting the neurons that react to a specific object appears to rely on some central resource [72, 73] and to spread gradually through the feature maps [34, 35, 63]. We used a computer vision algorithm for this step, which centrally computes the eigenvectors of the connectivity graph Laplacian [2], which does not immediately look biologically plausible. However, a recent theory for hippocampal place and grid cells suggests that these cells compute the same eigenvectors of a graph Laplacian of a prediction network, albeit of a successor representation, i.e. of predictions of the animals state transitions [68, 69]. Thus, this might be an abstract description of an operation brains are capable of. In particular, earlier accounts that model the selection as a marker that spreads to related locations [e.g. 17, 58, 67] have some similarities with iterative algorithms to compute eigenvectors. Originally, phase coherence between the neurons encoding the same object was proposed [17, 57, 67], but a gain increase with object based attention [58] or a known random modulation is also sufficient to select a task relevant set of neurons [25, 26]. Regardless of the mechanistic implementation of the marker, connectivity information of the type our model extracts would be extremely helpful to explain the gradual spread of object selection.

Our implementation of the model is not fully optimized, as it is meant as a proof of concept. In particular, we did not optimize the architectures or training parameters of our networks for the task, like initialization, optimization algorithm, learning rate, or regularization. Presumably, better performance in all benchmarks could be reached by adjusting any or all of these parameters.

One possible next step for our model would be to train deeper architectures, such that the features could be used for complex tasks like object detection and classification. Contrastive losses like the one we use here are successfully applied for such pretraining purposes even for large scale tasks such as ImageNet [60] or MS Coco [48]. These large scale applications often require modifications for better learning though [11, 15, 22, 28, 29, 75]. For example: Image augmentations to explicitly train networks to be invariant to some image changes, prediction heads that allow more complex shapes for the predictions, and memory banks or other methods to decrease the reliance on many negative samples. Similar modifications might be necessary to apply our formulation to deeper architectures for pretraining purposes. For understanding human vision, this line of reasoning opens the exciting possibility that higher visual cortex could be explained based on similar principles, as representations from contrastive learning also yield high predictive power for these cortices [86].

The model we propose here is a probabilistic model of the feature maps. One implication of this is that we could also infer the feature values if they were not fixed based on the input. Thus, our model implies a pattern how neurons should combine their bottom-up inputs with predictions from nearby other neurons, once we include some uncertainty for the bottom-up inputs. In particular, the combination ought to take into account which nearby neurons react to the same object and which ones do not. Investigating this pooling could provide insights and predictions for phenomena that are related to local averaging like crowding for example [3, 18, 30, 77–79], where summary statistic models currently capture perceptual limitations best [3, 18, 78], but deviations from these predictions suggest that object boundaries change processing [30, 77, 79].

Another promising extension of our model would be processing over time, because predictions over time were found to be a potent signal for contrastive learning [15] and because coherent object motion is among the strongest grouping signals for human observers [38] and computer vision systems [85]. Beside the substantial increases in processing capacity necessary to move to video processing instead of image processing, this step would require some extension of our framework to include object motion into the prediction. Nonetheless, including processing over time seems to be an interesting avenue for future research, especially because segmentation annotations for video are extremely expensive to collect such that unsupervised learning is particularly advantageous and popular in recent approaches [1, 33, 44].

This work aims to move us closer to understanding how human visual perception can take object structure into account in retinotopic feature map processing and may help us to build systems with similar capabilities in the future. We acknowledge that such technological progress can have unknown societal consequences, but we do not foresee specific negative consequences of this work.

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
