# OpenReview forum: "Unsupervised learning of features and object boundaries from local prediction"
_NeurIPS.cc/2022/Conference — NeurIPS 2022 Submitted_

### Official Review · Reviewer_6B8P · 2022-07-11

**Rating:** 8
**Confidence:** 4
**Soundness:** 4 excellent
**Presentation:** 3 good
**Contribution:** 3 good

**Summary:**

The authors propose an unsupervised learning method which learns local features from image pixels both with weights that allow to group those features together.

The contributions are the following :
- a description of their original model based on Markov Random Fields,
- explanations about how to trained the proposed models following principle from contrastive learning,
- a comparison of the learned features under variation of their algorithm (loss, linear/seg),
- a comparison of the performance of their algorithm on a contour detection task (BSD500).

**Questions:**

1. The issue with the position loss and the trick to circumvent it are obscure to me. Can you elaborate more on the issue and its solution ? I do not really understand the issue and how the proposed trick solves it.

2. I disagree with the fact that the linear models learn Gabor-like features. In particular for the position loss (Figure 3.A), the features are more like *non-local* wave-like features (Fourier).

3. I would like to see some maps of p_ij or w_ij associated to some features to understand how the algorithm has grouped those.

4. A related question is that the post-processing step to turn p_ij into contours makes hard to understand what was learned ? Is the use of spectral clustering transparent ie has the algorithm learned to group contours over any other type of grouping (eg textures) ?

Few additional references about texture-based segmentation :
- Beck, J., Sutter, A., & Ivry, R. (1987). Spatial frequency channels and perceptual grouping in texture segregation. Computer Vision, Graphics, and Image Processing, 37(2), 299-325.
- Landy, M. S., & Bergen, J. R. (1991). Texture segregation and orientation gradient. Vision research, 31(4), 679-691.
- Wolfson, S. S., & Landy, M. S. (1995). Discrimination of orientation-defined texture edges. Vision research, 35(20), 2863-2877.
- Vacher, J., Launay, C., & Coen-Cagli, R. (2022). Flexibly regularized mixture models and application to image segmentation. Neural Networks, 149, 107-123.

Typos :
l 63 -> which we can us to


**Limitations:**

Limitations are sufficiently assessed.

**Strengths And Weaknesses:**

Strengths :
- modelisation of grouping using a binary weight which is on or off whether the feature are grouped or not,
- a method to train a seemingly untrainable model (without supervision),
- multiple variations of their model are compared,
- model comparison with SOTA algorithms,
- a real concern about visual perception

Weaknesses :
- results are not SOTA,
- non-negligible post-processing to extract image contours
- learned features are shown but not the local weights given a feature and a location

---

> ### Author Response · Authors · 2022-08-02
> **Response to Reviewer (Part 1)**
>
> We thank the reviewer for their positive comments! For this reviewer, we think that the questions contain the weaknesses almost exactly. Thus, we answer only the questions here:
>
> ### Questions
>
> > The issue with the position loss and the trick to circumvent it are obscure to me. Can you elaborate more on the issue and its solution ? I do not really understand the issue and how the proposed trick solves it.
>
> The issue for the position loss is that the denominator for the contrastive loss (Eq. 8) is different for each position. Each position has a different neighborhood and thus requires a different normalization constant for the conditional distribution $p(f | f_{neigh})$. As we process all positions in parallel this incurs a very high memory load. Intermittently, the computation of the denominator generates a tensor of dimensions $N_{features} \times N_{neighbors} \times N_{negative samples} \times N_x \times N_y$. This severely limits the number of negative samples that we can use, which is not great, because contrastive methods require a decent number of negative samples to work well.
>
> Our trick to solve this problem is nothing complicated. We simply compute the loss multiple times with a small number of negative samples (usually 10), which is similar to using many negative samples, but avoids the memory explosion.
> When implementing this trick, we save a little computation by splitting the computation of the gradients at the feature map level. We split the computation into two steps: Image $\rightarrow$ feature map and feature map $\rightarrow$ loss. To compute multiple losses we run only the second part multiple times summing the gradients we get for the feature map. We then need only a single backpropagation through the computation of the feature map from the image.
>
> We hope this made this technical step clearer. We added some explanation to this step to our manuscript as well such that we confuse future readers less. In the manuscript we now write:
>
> “To enable a sufficiently large set of negative points $i'$ with the available memory, we compute this loss multiple times with few negative samples and sum the gradients. This trick saves memory, because we can free the memory for the loss computation after each repetition. As the initial computation of the feature maps is the same for all negative samples, we can save some computation for this procedure by computing the feature maps only once. To propagate the gradients through this single computation, we add up the gradients of the loss repetitions with regard to the feature maps and then propagate this summed gradient through the feature map computation. This procedure does not save computation time compared to the loss with many negative samples, as we still need to calculate the evaluation for each position and each sample in the normalization set.”
>
> > I disagree with the fact that the linear models learn Gabor-like features. In particular for the position loss (Figure 3.A), the features are more like non-local wave-like features (Fourier).
>
> We can certainly understand where the reviewer comes from. The features do not look like they smoothly taper down towards the edges of their filters. We would argue that they are nonetheless local features as their convolutional filters or receptive fields do end. Perhaps some of them are more akin to a boxcar filter times a wave than to a Gaussian times a wave, but they are spatial frequency and orientation specific filters with a limited spatial extent. We hope that the reviewer can agree with us on that.

---

> ### Author Response · Authors · 2022-08-02
> **Response to Reviewer (Part 2)**
>
>
> > I would like to see some maps of p_ij or w_ij associated to some features to understand how the algorithm has grouped those.
>
> We have looked at all those maps extensively during the development of our method. We usually favored the maps across all features, as our model assumes a single w which applies to all features. We computed these maps for the example images and the winning models and added them to the supplementary material. They take up quite a lot of space to show at a reasonable resolution, which is why we cannot add them to the manuscript.
>
> The individual maps generally look like oriented edge detector responses, where the orientation corresponds to the displacement between the neighbors. The raw probabilities for the w are somewhat noisier than the final contours we extract, but we hope that the reviewer agrees that the information is already present quite completely in the maps.
>
> Some informal observations on the relation between features and the MRF parameters reveal the expected relationships: Factors for closer neighbors have higher prior probability for connection $p$ and higher precision parameters in $C$. Local averages are connected in all directions, and the stripe-like patterns have stronger coupling along the stripes than orthogonal to them.
>
> > A related question is that the post-processing step to turn p_ij into contours makes hard to understand what was learned ? Is the use of spectral clustering transparent, i.e. has the algorithm learned to group contours over any other type of grouping (e.g. textures) ?
>
> For evaluation, we required an algorithm that turns local connectivity into contours, i.e. probable locations for object boundaries. This is nontrivial, because the local connections can be inconsistent, which makes this question a valid concern.
>
> However, we chose the most canonical algorithm for this transformation we could find. The globalization algorithm we use is provided with the BSDS500 dataset and has been applied to other edge features and segmentation algorithms previously as discussed in detail in the paper by the authors of the BSDS500 dataset (Arbeláez et al., 2011). Thus, we believe this algorithm is among the most transparent ones available for the transformation we require and that we can trust that the contours found by the algorithm match the connectivity information we input well.
>
> Conversely, this algorithm does not automatically lead to good performance. Arbeláez et al. (2011) applied the algorithm to other local edge detection signals, too. Typically, this led to some improvement, but the simple edge detectors still performed badly. Similarly, different feature extraction networks yield substantially different performance in our evaluations, indicating that the features and connectivities are important.
>
> Finally, we now provide the raw inferred probabilities for the example images in the supplementary material. We found the direct comparison of a simple sum of connectivities and the inferred contours quite convincing that the main contribution comes from the inferred features and connectivities, not from the globalization algorithm.
>
> **References:**  We included the given references into our manuscript. Thank you for pointing them out!

---

> > ### Comment · Reviewer_6B8P · 2022-08-04
> > **More information ?**
> >
> > Thanks to the authors for their answers.
> > After reading other reviews et responses I am still very positive about this manuscript. In particular, I hope that the response to reviewer NeiE will convince him to raise his score !
> >
> > I thank the author for including w_ij maps in the supplementary. However, I am a bit unsure how to read those figures. What are the two images in the files number.pdf ? What are the 10 b&w images in the files number_w_map.pdf. I guess those are the w_ij maps but I would have expected that such a map is given for a fixed index ie set i_0 and show w_i0j. Here I don't see any reference pixel for each map...
> >
> > If I understood correctly the weight w_ij tells us whether the pair of features f_i and f_j is active (ie whether the pixels i and j should be grouped together, right ?). Then, in the results, the w_map you are showing seem to be only active along contours. Does it mean that your algorithm is only learning to group contours ? I would have expected to see some texture grouping.
> >
> > This was a bit crowded among other questions in 4. but I would have appreciated to see a discussion about the type of grouping that the algorithm has performed ie contour vs texture grouping. Could you elaborate on that and discuss it in the paper ?

---

> > > ### Author Response · Authors · 2022-08-05
> > > **More Information !**
> > >
> > >
> > > Thank you again for your positive evaluation!
> > >
> > > Sorry, that we did not properly explain the new figures. The *w_maps* we display are the log-posterior ratios for all the $w$s being 1 vs being 0 for the given image. To display these, we put all values that correspond to the same spatial relationship between the neighbors into one image. This is what is displayed in the `number_w_map.pdf` files. For example, the first map shows this log-posterior ratio for the connection from each pixel to their right neighbor. The second one shows the connection to the pixel below. The third one is for 1 pixel to the right and one pixel down, etc. There are 10 maps here for the 20 neighborhood relations, because our factors are symmetric. Thus, each map is exactly equal to the map for the opposite shift between pixels, e.g. the map is the same for one pixel up as for one pixel down. We display only one of these each, which halves the number of maps.
> > >
> > > The colormap is the other way around than you assume (black is active, white is inactive). The w_maps are active everywhere but at the contours, i.e. most pixels should be grouped with their close neighbors, only a few along the contours should not be.
> > >
> > > In the files named `number.pdf`, the second panel is a simple sum of the w_maps for the different neighborhood relationships (all 20 in this case). This also creates an approximation of the contours without any graph spectral methods. We hope the comparison to the contours returned by the globalization algorithm in the third panel is convincing that globalization is not the main driver of algorithm performance.
> > >
> > > ---
> > >
> > > Last but not least on the texture vs. contour grouping distinction: We think of our method mostly as texture grouping: We extract (texture or surface) features at every location and closeby locations with similar features are grouped together, while locations with different features are not. In particular, none of our filters are directly used as contour detectors. The globalization algorithm is just a convenient way to turn this similarity information into contours in a way that is strongly related to graph spectral clustering.
> > >
> > > This classification is not perfectly clear though. Someone else might argue that our algorithm instead uses the feature maps only as an intermediate step to calculating a complicated edge detector that returns the w_maps as the contours. These contours are then grouped by the globalization algorithm. This highlights to us that these two interpretations are truly aspects of the same problem.
> > >
> > > We will try to find space to say something about this distinction in the manuscript.

---

> > > > ### Comment · Reviewer_6B8P · 2022-08-09
> > > > **thanks**
> > > >
> > > > Thank you for your answer this is much clearer now !
> > > > I really hope that you'll find the space to include the discussion about texture vs contour grouping (I could definitely find something to remove but I am not author, this is your decision).
> > > > I am happy to increase my score and I hope to see this paper accepted !

---

> > > > > ### Author Response · Authors · 2022-08-09
> > > > > **Thank you!**
> > > > >
> > > > > Thank you! We are glad that we could clarify things and hope to see this paper accepted, too, of course.
> > > > >
> > > > > Also, If our submission is accepted, we will be allowed an additional content page for the camera-ready version. Thus, we will most certainly have space to add a paragraph on texture vs. contour grouping.

---

### Official Review · Reviewer_TYhZ · 2022-07-11

**Rating:** 7
**Confidence:** 3
**Soundness:** 3 good
**Presentation:** 3 good
**Contribution:** 3 good

**Summary:**

* In this work the authors propose a model that can learn visual input features and segmentation without supervision which combines contrastive learning + Markov Random Fields (MRFs) to implement spatial feature prediction for feature maps + image segmentation.
* Self supervised learning approach: prediction between locations yield self supervised loss to learn feature maps, how to infer which locations should be grouped.
* Gaussian Markov Random Fields are used to model pairwise connections among features given feature maps and binary variables that indicate activity between connections.
* When considering pairwise connections the four adjacent pixels from the neighbours of position i. The factors of features (point or pairwise) model the strength and influence of a point on its neighbours wrt to a feature.
* There are two contrastive Learning Objectives defined on the marginal likelihood p(f)  1. Position Loss: optimizes probability of randomly chosen positions from other features + images 2. Factor Loss: maximizes factor for correct feature vectors relative to random feature vectors sampled from different locations and images.  Optimization is done on all parameters of the MRF in parallel via SGD with momentum
* Segmentation Inference: p(w_i,j = 1 | f) is the probability that two points are connected.  This depends on the two feature vectors f_i ^& f_j.  This is computed by forming a sparse connectivity matrix from: p(w_i,j = 1 | f) / p(w_i,j = 0 | f) (claimed to be an attention dependent serial process)
* Features are trained over three Models: Identity, Linear, and ResNet. Models are trained on MS COCO image dataset. The authors provide an analysis of the learned features for the linear and ResNet model.  Contours are extracted from the model and compared to the Berkeley (human) Segmentation Database (BSDS500).  The authors present recall and precision scores along with other segmentation metrics against a number of other baselines that may have been trained with additional supervision signals ( human contour / segmentation data + pretraining on other tasks).

Main claims of the work:

1. Predictions across space aid in segmentation and feature learning without further supervision signals (human contour data or pre-training on other tasks/data).
2. Models optimized under this approach have similarities to the human visual system (retinotopic visual cortex)


**Questions:**


It was stated that models using the factor loss + neighbourhood of size 20 performed best.  Was there a sense of whether further increases in the neighbourhood would yield further improvement in the scores?  Were any other losses considered?

In section 3.2 it's stated: "... the best performing models seem to be mostly the local averaging models ..." Do you have any intuition or explanation as to why the best performing models were those that do local averaging?

Could this approach induce good representations for image or video data for downstream tasks or other applications?  Was this explored at all?

Would prediction across features in the manner presented be applicable to other input modalities (e.g. language/text) or sensory data (e.g. speech or audio)?


**Limitations:**

The authors discuss the limits of their model in the context of low scale and minimal tuning - in effect this is a proof of concept.  They also note that the contrastive losses + learning are not optimal and could be modified to achieve better optimization.  Further the model does include any temporal signal which may also aid in helping to better model probabilistic features.

**Strengths And Weaknesses:**

**Strengths**

* Overall I believe this is a pretty interesting paper although I'm not deeply familiar with all of the ties to cognitive science and the human visual system.  However, I think that the authors do a good job of demonstrating links to the human visual system via the contour & segmentation analysis and the feature visualization without any specific supervision toward human visual cognition.
* The results are clearly demonstrated and show that even with simple feature modeling that spatial feature prediction modeled by the MRF + contrastive losses the model achieves reasonable segmentation and contour prediction accuracy approaching the level of deep neural networks making use of pretraining on other tasks and training on human reported contours.
* In the discussion there are some highlights regarding how this method performs object selection and how temporal data might be used to further improve model capacity to detect object structure (the binding problem).  Also the discussion of biological plausibility helps to underpin the relevance of this work.
* The scores achieved look fairly good despite relatively little tuning and relatively small models even though optimizing over these metrics wasn't the aim of this research.

**Weaknesses**

* May need more analysis/explanation on the reasons behind why the factor loss performed well vs. the position loss.
* Sub-section 2.3 may be more appropriately placed under section 3.  Are these meant to be baselines? Finally, Fig. 2 appears fairly early in the text relative to this subsection.
* The authors stated a number of future directions for this work in section 5 including processing video input.  It would be helpful to know if we could reasonably expect these models to advance state of the art and how this approach fits into the broader family of models tackling image/video segmentation and representation.
* Figure 2 D & E were a bit tricky to understand and might benefit from some clarification in the description or main text.

---

> ### Author Response · Authors · 2022-08-02
> **Response to Reviewer (Weaknesses)**
>
> We thank the reviewer for their positive comments and answer to their questions and mentioned weaknesses below, starting with the weaknesses:
>
> ### Weaknesses
>
> > May need more analysis/explanation on the reasons behind why the factor loss performed well vs. the position loss.
>
> We were not surprised by the factor loss performing better than the position loss. We believe the main reason for the factor loss performing well is the higher efficiency due to reusing the negative samples across positions. This approach allows us to use a much larger negative sample size for the factor loss ([batch size] * [image size] \approx 100000 for the factor loss vs. 50 or 100 for the position loss). This leads to much faster convergence of the training and the resulting features look much smoother for the factor loss. Thus, we believe that the advantage for the factor loss is simply due to the higher technical efficiency. We now state this belief in the manuscript when we note the higher performance.
>
> > Sub-section 2.3 may be more appropriately placed under section 3. Are these meant to be baselines? Finally, Fig. 2 appears fairly early in the text relative to this subsection.
>
> We did not have strong opinions on these placements. We moved the start of section 3 earlier and Fig. 2 is now gone. We hope this makes our manuscript better.
>
> > The authors stated a number of future directions for this work in section 5 including processing video input. It would be helpful to know if we could reasonably expect these models to advance state of the art and how this approach fits into the broader family of models tackling image/video segmentation and representation.
>
> We do think that our approach could lead to state of the art representations learning in the future and that processing over time is indeed one of the most likely areas where this could happen. We state this now in our discussion using the space gained by removing Fig. 2. We do not want to phrase this too strongly, as we implemented neither processing across time, nor evaluated our models on these tasks.
>
> > Figure 2 D & E were a bit tricky to understand and might benefit from some clarification in the description or main text.
>
> We thought that we do not have the space to explain this figure sufficiently and thus removed it. We hope that our added explanations made the paper clearer then the figure did.

---

> > ### Comment · Reviewer_TYhZ · 2022-08-08
> > **Response to Authors**
> >
> > Thanks for these clarifications and the modifications to the manuscript.  In particular I appreciate the discussion around feature learning, I'd be curious to see how this approach fares as a pre-training step for image encoders for downstream tasks.

---

> ### Author Response · Authors · 2022-08-02
> **Response to Reviewer (Questions)**
>
> ### Questions
>
> > It was stated that models using the factor loss + neighborhood of size 20 performed best. Was there a sense of whether further increases in the neighborhood would yield further improvement in the scores? Were any other losses considered?
>
> We think that larger neighborhoods would work even better given that performance increases monotonically with the neighborhood size and very large neighborhoods or even fully connected MRFs work better in similar semantic segmentation tasks. We added this point to the manuscript as follows:
>
> “Performance increases monotonically with neighborhood size and Markov random field based approaches to semantic segmentation also increased their performance with larger neighborhoods up to fully connected Markov random fields \cite{krahenbuhl2012, chen2014, chen2017}. We thus expect that larger neighborhoods could work even better.”
>
> We did not consider any other losses beyond the ones in the paper.
>
> > In section 3.2 it's stated: "... the best performing models seem to be mostly the local averaging models ..." Do you have any intuition or explanation as to why the best performing models were those that do local averaging?
>
> This observation holds primarily in contrast to other, more Gabor-like features. Our intuition is that the models that learn more gabor-like features tend to classify some contours as separate objects with boundaries on either side of them (Some effects in this direction can be seen by zooming into the example images for the linear model and layer 1 of predseg1). This leads to lower performance on the benchmark. However, this argument contains a lot of intuition and we would like to find some kind of formal evidence for this being the difference before making strong claims about this in our paper.
>
> > Could this approach induce good representations for image or video data for downstream tasks or other applications? Was this explored at all?
>
> We hope that this would be the case and both the success in finding good low-level features and the success of contrastive learning as a pretraining scheme for other tasks lends support to this idea. We did not explore this with our model so far, because our architectures were quite shallow and we thus do not expect that the features we find are sufficient for object classification or similarly complex tasks. We now reference this with our future directions in our rewritten paragraph on training deeper architectures:
>
> “One possible next step for our model would be to train deeper architectures, such that the features could be used for complex tasks like object detection and classification. Contrastive losses like the one we use here are successfully applied for such pretraining purposes even for large scale tasks such as ImageNet \cite{russakovsky2015} or MS Coco \cite{lin2015}. These large scale applications often require modifications for better learning though \cite{chen2020, feichtenhofer2021, grill2020, he2020, henaff2020, oord2019}. For example: Image augmentations to explicitly train networks to be invariant to some image changes, prediction heads that allow more complex shapes for the predictions, and memory banks or other methods to decrease the reliance on many negative samples. Similar modifications might be necessary to apply our formulation to deeper architectures for pretraining purposes. For understanding human vision, this line of reasoning opens the exciting possibility that higher visual cortex could be explained based on similar principles, as representations from contrastive learning also yield high predictive power for these cortices \cite{zhuang2021}.”
>
> > Would prediction across features in the manner presented be applicable to other input modalities (e.g. language/text) or sensory data (e.g. speech or audio)?
>
> We do think that the ideas we present here could generalize to other modalities. For example, one could mix the predictions of a text or speech prediction system with a broader uninformed distribution as we do here for predictions across space in an image. One could then use the relative prediction of these two cases for the segmentation of the stream into syllables, words, etc., analogous to our image segmentation. There was enough to say about the visual modality, such that we did not find space to discuss this in the paper, but it would be an interesting application to look at in the future.

---

### Official Review · Reviewer_NeiE · 2022-07-12

**Rating:** 3
**Confidence:** 4
**Soundness:** 3 good
**Presentation:** 2 fair
**Contribution:** 2 fair

**Summary:**

The paper proposes an unsupervised model for segmentation and contour detection. The unsupervised learning is based on training Markov Random Field model find new feature maps. The MRF is constructed by connectivity of pixels in a local image patch, and Gaussian responses (called factors). The feature maps are inserted into shallow neural networks, and ideal features are learned via contrastive loss based on noisy and correct locations of the pixels and gaussian factors. as well as target which can then be inserted into forward DNNs.

**Questions:**

See above.

Minor:
Line 3: .”..dealt with separately,” -> “ dealt separately” ?


**Limitations:**

yes

**Strengths And Weaknesses:**

Strengths:
•	I agree with the authors that more research on grouping mechanisms for neural network models is valuable and can contribute to improved CNN for downstream visual recognition tasks.

Weaknesses:
•	The paper is hard to follow, and more intuitive explanations on the mathematical derivations are needed. Figure captions are lacking, and require additional explanations and legends (e.g., explain the colors in Fig. 2). Fig. 1 and 2 did not contribute much to my understanding, and I had to read the text few times instead.
•	At the end of the day the model proposes a method to learn features for detecting boundaries, which is an old computer vision task. Indeed, it uses a new MRF framework, and contrastive loss, but it is not clear why not using DNNs with contrastive loss for doing that, besides that maybe the learned features are more like human vision features.
•	The results of the models are not compared against unsupervised DNN models. I think it is interesting to see such a comparison, e.g., to unsupervised segmentation models that can be adjusted to the BSDS500 contour detection task.
•	There is not review of previous related work, and these is not section for “related work”. Can unsupervised segmentation model be relevant to your work?

---

> ### Author Response · Authors · 2022-08-02
> **Response to Reviewer**
>
> Unfortunately, it seems that we did not get some of our central points across. To prevent such misunderstandings for future readers, we removed Figure 2, which was not explained sufficiently and added some clarifying sentences to the explanation of our model to make the mathematics more intuitive. Furthermore, we added a few comparison points from the deep neural networks for segmentation, although there are very few truly unsupervised models as most models require some form of supervision like human reported contours or segments.
>
> Additionally, we want to clarify two points here:
>
> First, we believe that there was some misunderstanding regarding the structure of our model. The features we compute are not “connectivity of pixels in a local image patch”, but filters of a shallow convolutional neural network which computes the maps, which we model with the markov random field model. In other words: We first compute these feature maps with a DNN architecture and model the output feature maps using a MRF.
> This is important, because it means that our models are deep neural networks trained with contrastive learning, just as the reviewer asks for as a potentially better alternative. Also, this means that our approach is applicable to much deeper architectures for computing the feature maps than we used here and definitely different from the classical computer vision approaches.
>
> Second, we did reference alternative approaches in our introduction and discussion already, just not in a separate section on related work. We happily included a few references to unsupervised learning for segmentation in deep neural networks there, together with literature the other reviewers mentioned. We want to emphasize that approaches that are truly unsupervised, i.e. do not use any segmented data for training, are rare. Most references we found that follow the approach of “unsupervised segmentation models that can be adjusted to the BSBS500 contour detection task” as suggested by the reviewer require human contours to train the adjustment and read out mechanisms.
>
> We hope that these comments improve clarity for both the reviewer and future readers and that the reviewer might reconsider their recommendation for rejection.

---

### Author Response · Authors · 2022-08-02
**General response to reviewers**

We thank the reviewers for their insightful comments and the overall positive evaluation of our manuscript! It seems that the one reviewer’s negative evaluation is largely due to some misunderstandings, which we hope our responses will rectify.

We were particularly happy that the reviewers highlighted that we “do a good job of demonstrating links to the human visual system” and that we have “a real concern about visual perception”, as understanding human vision is our final goal. Additionally, we appreciate that the reviewers evaluate the performance of our model and paper positively despite not being SOTA, because achieving SOTA in our case would most likely involve a level of tuning that is incompatible with the claim that the model learns without supervision.

Most of the concerns were about some aspects of our model presentation being unclear and additional reference points we should relate our model to. To address these points, we decided to remove Figure 2 from our manuscript as two reviewers felt it was insufficiently explained to be useful. We had included it, because it was a helpful illustration in presentations we gave on this work, but for the written article it seems that we cannot give sufficient context to make it useful.

Removing the figure allowed us to discuss most points the reviewers raised in the manuscript as well. In particular, we are now able to give some additional clarification on the differences we observe between the position and factor losses, add some discussion on previous unsupervised deep neural network models and previous work on texture segmentation, and some more hints to prevent misunderstandings of our architecture as we believe happened for one of the reviewers.

We believe these changes improve the clarity of our submission and answer the reviewers comments as we discuss in direct replies to each of them.

---

### Meta-Review · Area_Chair_rEcp · 2022-08-25

**Recommendation:** Reject
**Confidence:** Certain

**Metareview:**

There was some disagreement on the value of this work. The paper received 1 strong accept, 1 accept and 1 reject. The positive reviewers recommended the paper to be accepted because it proposes a novel unsupervised approach to semantic segmentation and contours detection and because of connections to neuroscience. Some of the main criticisms from the more negative reviewer included a lack of a discussion to related work and a lack of sufficient experimental evaluation (in particular comparisons to related work).

The AC found the response of the authors in the rebuttal unconvincing. There is quite a bit of prior work using MRF for segmentation (yes prior supervised segmentation work needs to be properly cited and not just in passing in the results section (line 216-217). Wrt a lack lack of baseline comparisons, I also agree with the reviewer and in that respect the statement in the paper (line 236-237) is not convincing ("There are also a few deep neural network models that attempt unsupervised segmentation [e.g. 10, 47, 82], but we were unable to find any that were evaluated on the contour task of BSD500"). It seems that the authors should at least consider running these models on BSD500 or run their models on other datasets. And as acknowledged by one of the more positive reviewers the results are not SOTA. As stated in the discussion, the authors plan to continue tweaking the architecture to improve results. The AC thinks this is needed for this work to make a sufficient contribution to the conference since neither the use of MRFs or contrastive loss are novel on their own -- the burden is on the authors to demonstrate that they can engineer a system from these two ideas with at least competitive results.

As for the neuroscience contribution, I am quoting the discussion ("the features learned by our models resemble receptive fields in the retina and primary visual cortex and the contours we extract from connectivity information match contours drawn by human subject fairly well, both without any training towards making them more human-like.").  This seems like an unsubstantiated statement since no quantitative analysis is provided. Almost any CNN trained for any reasonable task will return some sort of center-surround, orientation-selective and color-opponency filters (take AlexNet trained on ImageNet for instance) -- so it is unclear what is new here. For this statement to be meaningful that authors should formulate null models and demonstrate empirically that their proposed model are more cortex-like than other reasonable alternative model.

Overall because of a lack of sufficient technical or neuroscience contribution, the AC recommends this paper to be rejected.

**Award:**

No

---

### Decision · Program_Chairs · 2022-09-14

Reject